# Cancer stage at diagnosis by duration of pre-existing chronic analgesic use and anxiety or depression

Helen Fowler [1] ✉, Georgios Lyratzopoulos [1], Meena Rafiq[1,2], Matthew E. Barclay [1], Gary A. Abel[3] & Cristina Renzi[1,4]

Pre-existing chronic diseases may delay or expedite cancer diagnosis. Here, we examine variations in cancer stage at diagnosis based on duration and type of common chronic conditions. We identify lung and colon cancers diagnosed 2012-2018 from national cancer registration, and pre-existing physical and mental-health conditions from linked primary care records. Using multi-variable logistic regression, we explore associations between the most pre-valent conditions (Anxiety/Depression and Chronic Analgesic Medication use), classified as "Recent-onset" (first recorded <12months pre-cancer) or "Persistent/Historic" (12-72 months pre-cancer), and cancer stage at diagnosis. We show that recent-onset Analgesic Medication use can be associated with increased odds of advanced stage lung or colon cancer diagnosis. Conversely, persistent/historic Chronic Analgesic Medication use can be associated with reduced odds of advanced stage lung cancer and persistent/historic Anxiety/Depression with reduced odds of advanced stage lung or colon cancer. Persistent or historic conditions may increase healthcare utilisation, offering opportunities for early cancer diagnosis. Recent-onset conditions may lead to delays through the alternative explanations or competing demands mechanisms.

At the time of cancer diagnosis many patients are already living with one or more chronic conditions, also known as comorbidities[1]. The presence of pre-existing conditions can influence diagnostic pathways and the stage at which cancer is diagnosed. Diagnosis can either be expedited or delayed, depending upon different underlying mechanisms that may impact the diagnostic process. For example, patients with pre-existing conditions may consult with their healthcare providers more frequently, leading to increased opportunities for reporting and subsequent investigation of potential cancer symptoms and an earlier cancer diagnosis through a 'surveillance' mechanism. Conversely, chronic conditions may delay cancer diagnosis where the patient and/or healthcare professional prioritise managing the chronic condition rather than investigating new and possibly vague symptoms

through a 'competing demands' mechanism, or when symptoms of an as-yet undiagnosed cancer are attributed to the existing condition through an 'alternative explanation' mechanism[2,3].

Some studies report that common physical conditions or mental health conditions such as Anxiety or Depression can be associated with longer time to diagnosis, higher odds of emergency presentation and advanced stage cancer diagnosis[2,4]. However, the evidence is mixed[5], and the magnitude and direction of associations can differ according to the cancer site as well as the type and timing of onset of different chronic conditions[6]. The interpretation of findings is challenged by inconsistencies in the definitions and look-back period for identifying chronic conditions[5]. Additionally, the impact of treatment for pre-existing chronic conditions, for example opioid treatment for chronic

[1]Epidemiology of Cancer Healthcare and Outcomes Group, University College London, London, UK. [2]Centre for Cancer Research and Department of General Practice, University of Melbourne, Melbourne, Australia. [3]Department of Health and Community Sciences, Faculty of Health and Life Sciences, University of Exeter Medical School, Exeter, UK. [4]University Vita Salute San Raffaele, Milan, Italy. ✉e-mail: h.fowler@ucl.ac.uk

pain, may influence the interpretation of cancer symptoms, and ultimately impact timing of cancer diagnosis. Similarly, use of several different prescription medications for the treatment of pre-existing conditions may also complicate the interpretation of cancer symptoms. A population-based Danish study of twenty cancers found that one-third of patients had received prescriptions for 5 or more medications during the year before their cancer diagnosis[7]. The presence of pre-existing conditions may also increase the likelihood of patients presenting to their GP with non-specific symptoms. Routine blood testing to identify the cause of the symptoms may lead to cancer being detected following further investigation of abnormal blood results. A Danish study investigating the probability of a cancer diagnosis among a group of patients undergoing routine blood tests for non-specific serious symptoms found that patients with specific combinations of two abnormal blood tests had up to 62% probability of having cancer[8]. While the impact on diagnostic pathways and cancer stage might vary depending on the timing and duration of chronic conditions, typically studies have not distinguished between long-standing or recent-onset conditions[6].

In our study we chose to focus on lung and colon cancers. These are common cancers[9], frequently diagnosed at advanced stage[10] and via emergency presentation[11], and having high mortality[12]. Our study had two objectives. Firstly, we used primary care records to identify the prevalence of thirty-five pre-existing conditions, according to timing of onset and duration, among patients diagnosed with lung or colon cancer. Secondly, we investigated associations between the most common pre-existing physical and mental health conditions and stage at lung or colon cancer diagnosis, accounting for the duration of these chronic conditions.

In this work we show that Anxiety or Depression and Chronic Analgesic Medication use are common pre-existing conditions among patients diagnosed with lung or colon cancer. We show that recent-onset Chronic Analgesic Medication use can be associated with increased odds of advanced stage lung or colon cancer diagnosis. In contrast, persistent/historic Chronic Analgesic Medication use can be associated with reduced odds of advanced stage lung cancer and persistent/historic Anxiety or Depression can be associated with reduced odds of advanced stage lung or colon cancer.

Persistent/historic conditions may lead to increased healthcare utilisation, offering opportunities for early cancer diagnosis. Recent-onset conditions may lead to delays in cancer diagnosis through the alternative explanations or competing demands mechanisms.

## Results

We identified 13,444 incident lung cancers and 8,345 incident colon cancers diagnosed 2012-2018 from England registry data linked with primary care records. From these records, we excluded patients i) without at least one GP record and ii) who were not registered at a CPRD Up To Standard GP Practice in the 12 months before diagnosis. Our final study population included 6,828 lung cancer and 4194 colon patients (Supplementary Fig. 1).

### Prevalence of physical and mental health comorbidities

Among the 6828 lung cancer patients, the most prevalent conditions were Painful Conditions (39%), Anxiety/Depression (26%), Chronic Obstructive Pulmonary Disease (COPD) (25%), Hypertension (11%), and Diabetes (10%) (Fig. 1, Supplementary dataset 1). Over a quarter of patients (27%) had recent-onset Chronic Analgesic Medication use, while 9% had persistent Chronic Analgesic Medication use. The prevalence of Anxiety/Depression did not vary by timing of onset (~ 8% patients in each group). Around three-quarters of lung cancer patients with either recent-onset Anxiety/Depression or recent-onset Chronic Analgesic Medication use had an advanced stage cancer diagnosis (73% and 79% of patients, respectively), compared with approximately half of patients with persistent conditions (56% and 49%, respectively) (Supplementary Fig. 2).

The most prevalent conditions among the 4,194 colon cancer patients were Chronic Analgesic Medication use (30%), Anxiety/Depression (21%), Hypertension (11%), Diabetes (10%) and Diverticular disease of the intestine (7%). More patients had recent-onset than persistent Chronic Analgesic Medication use (14% and 10% of patients, respectively,) while the opposite was true among patients with Anxiety/Depression (4% and 8% of patients, respectively). Most patients with either recent-onset Anxiety/Depression or recent-onset Chronic Analgesic Medication use had an advanced stage diagnosis (54% and 64% of patients, respectively) while less than half (40%) of patients with persistent conditions were diagnosed at advanced stage.

After grouping pain medication prescriptions by drug type (Supplementary Table 1), approximately half of the prescriptions of patients with Chronic Analgesic Medication use were for opioids, regardless of whether the conditions were recent-onset or persistent (Supplementary Fig. 3).

### Chronic analgesic use and anxiety/depression as predictors of stage

Patients with recent-onset Chronic Analgesic Medication use versus those without Chronic Analgesic Medication use had 63% increased

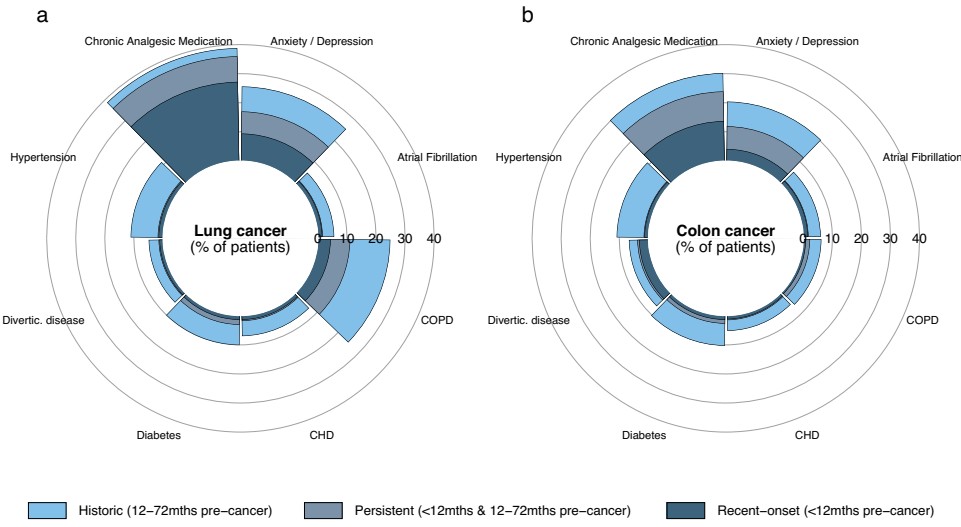

**Fig. 1 | Percentage (%) of lung and colon cancer patients with the top eight most prevalent conditions, by duration of condition. a** Lung cancer patients (*n* = 6828). **b** Colon cancer patients (*n* = 4194). *COPD* Chronic Obstructive Pulmonary Disease, *CHD* Congestive Heart Disease; Divertic. disease, Diverticular Disease of the Intestine.

odds of advanced stage lung cancer diagnosis, and more than doubled odds of advanced stage colon cancer (aOR 1.63; 95%CI 1.39, 1.91, and aOR 2.25; 95%CI 1.81, 2.79, respectively), adjusting for patient characteristics and primary care consultations (Table 1 and Fig. 2, Supplementary Table 2). By contrast, persistent/historic Chronic Analgesic Medication use was associated with lower adjusted odds of advanced stage, compared with no Chronic Analgesic Medication use, especially in the case of lung cancer (aOR 0.39; 95%CI 0.32, 0.47); the association for colon cancer was not statistically significant (aOR 0.96; 95%CI 0.78, 1.18). Similar associations were found in our sensitivity analysis, modelling stage as early (stages I-III) and advanced (stage IV)(Supplementary dataset 2).

**Table 1 | Odds Ratios (with 95% CIs) of advanced versus early-stage lung cancer diagnosis, adjusted for Anxiety/Depression, Chronic Analgesic Medication use and covariates**

|  | Lung cancer OR (95% CI) | Colon cancer OR (95% CI) |
|---|---|---|
| **Anxiety/Depression** | | |
| None (Ref.) | 1.00 | 1.00 |
| Recent-onset (<12 months pre-cancer) | 1.19 (0.94, 1.51) | 1.02 (0.71, 1.46) |
| Historic or Persistent (first recorded >12 months pre-cancer) | 0.83 (0.70, 0.98) | 0.77 (0.64, 0.93) |
| **Chronic Analgesic Medication use** | | |
| None (Ref.) | 1.00 | 1.00 |
| Recent-onset (<12 months pre-cancer) | 1.63 (1.39, 1.91) | 2.25 (1.81, 2.79) |
| Historic or Persistent (first recorded >12 months pre-cancer) | 0.39 (0.32, 0.47) | 0.96 (0.78, 1.18) |
| **Number of GP consultations (30 days to 2 years pre-cancer)** | | |
| 0-9 (Ref.) | 1.00 | 1.00 |
| 10-29 | 0.77 (0.60, 0.99) | 0.86 (0.70, 1.06) |
| 30+ | 0.55 (0.42, 0.71) | 0.88 (0.69, 1.12) |
| **Total number of physical comorbidities** | | |
| 0 | 1.00 | 1.00 |
| 1 | 0.95 (0.80, 1.13) | 0.99 (0.83, 1.17) |
| 2 | 0.93 (0.77, 1.13) | 0.92 (0.74, 1.14) |
| 3+ | 0.80 (0.65, 0.99) | 0.74 (0.58, 0.96) |
| **Haemoptysis** | | |
| No (Ref.) | 1.00 | – |
| Yes | 0.90 (0.63, 1.28) | – |
| **Cough or Dyspnoea** | | |
| No (Ref.) | 1.00 | – |
| Yes | 0.78 (0.69, 0.89) | – |
| **Fatigue or Weight Loss** | | |
| No (Ref.) | 1.00 | – |
| Yes | 0.99 (0.82, 1.21) | – |
| **Red-flag[a] symptoms** | | |
| No (Ref.) | – | 1.00 |
| Yes | – | 0.87 (0.70, 1.10) |
| **Non-red-flag[b] symptoms** | | |
| No (Ref.) | – | 1.00 |
| Yes | – | 1.25 (1.08, 1.46) |
| **Anaemia (blood test recorded)** | | |
| No (Ref.) | 1.00 | 1.00 |
| Yes | 0.98 (0.84, 1.14) | 0.97 (0.83, 1.14) |

Adjusted for listed covariates plus socio-demographic covariates; results for socio-demographic covariates are reported separately in Supplementary Table 2.
[a]Red-flag symptoms: Change in bowel habits and rectal bleeding.
[b]Non-red flag symptoms: abdominal pain/bloating, constipation or diarrhoea.

There was evidence of interaction between Chronic Analgesic Medication use and age (p < 0.01) (but not sex), among both lung and colon cancer patients. In lung cancer, for example, patients aged 60–69 years had 2.6 times the odds of advanced stage cancer (aOR 2.60; 95%CI 1.89, 3.57) if they had recent-onset Chronic Analgesic Medication use (vs none), but such patients aged 80+ years had 1.31 times the odds of advanced stage cancer vs a patient aged 60-69 without Chronic Analgesic Medication use (aOR 1.31; 95%CI 0.97, 1.76) – similar to the difference associated with such advanced age and no Chronic Analgesic Medication use (aOR 1.28; 95%CI 1.03, 1.60) (Supplementary Table 3). Among colon cancer patients, patients aged <60 years with persistent or historic Chronic Analgesic Medication use had 1.57 times the odds of advanced stage cancer versus patients aged 60-69 years with none (aOR 1.57; 95%CI 1.01, 2.43), while patients aged 80+ years with persistent or historic Chronic Analgesic Medication use had around half the odds of advanced stage cancer versus 60-69 year old patients with none (aOR 0.59; 95%CI 0.40, 0.89). Patients with persistent or historic Anxiety/Depression, compared to those without, had lower odds of advanced stage lung (aOR 0.83; 95%CI 0.70, 0.98) or colon (aOR 0.77; 95%CI 0.64, 0.93) cancer diagnosis (Table 1). Recent-onset Anxiety/Depression was not associated with stage of diagnosis of lung or colon cancer (aOR: 1.19; 95%CI 0.94, 1.51 and aOR: 1.02; 95%CI 0.71, 1.46, respectively). There was no evidence of an interaction between Anxiety/Depression and age or sex.

We did not find significant interaction between Anxiety/Depression and Chronic Analgesic Medication use (p < 0.05) (Supplementary Table 4).

From our sensitivity analysis modelling of Anxiety and Depression as separate conditions, only recent-onset Anxiety was associated with increased odds of advanced stage lung or colon cancer, while only historic or persistent Depression was associated with reduced odds of advanced stage cancer (Supplementary Table 5). In the main analysis, persistent Anxiety (combined with or without Depression) was associated with reduced odds of advanced stage cancer, but there was no association between recent-onset Anxiety/Depression and lung or colon cancer stage at diagnosis.

## Discussion

Chronic Analgesic Medication use and Anxiety/Depression were the most common pre-existing conditions, with at least 30% of patients with either lung or colon cancer affected by the former and 20% of patients by the latter. Our findings indicate that the association between these comorbidities and cancer stage at diagnosis is influenced by the timing and duration of the non-neoplastic conditions. Patients with recent-onset conditions (first recorded <12 months pre-cancer) had increased odds of advanced stage diagnosis, while those with persistent conditions (recorded both <12 months and 12-72 months pre-cancer) had reduced odds of an advanced stage diagnosis, compared to patients without the condition. The strength of these associations varied between the two cancer sites, although recent-onset Chronic Analgesic Medication use was strongly associated with advanced stage diagnosis of both lung and colon cancer.

There are a limited number of published studies investigating cancer stage at diagnosis among patients with pre-existing mental health conditions such as Anxiety or Depression[5]. One study found that depression is not associated with breast or colorectal cancer stage at diagnosis[13], while another found it to be associated with advanced stage pancreatic cancer[14]. There is also mixed evidence in the scientific literature on the associations between other pre-existing comorbidities and cancer stage at diagnosis. The magnitude and direction of these associations can vary depending on condition studied and cancer site, as highlighted in a recent systematic review of studies investigating stage at lung, breast, colorectal or prostate cancer diagnosis[6]. For example, a history of heart failure was associated with increased odds of advanced stage lung cancer in one study[15] but was associated

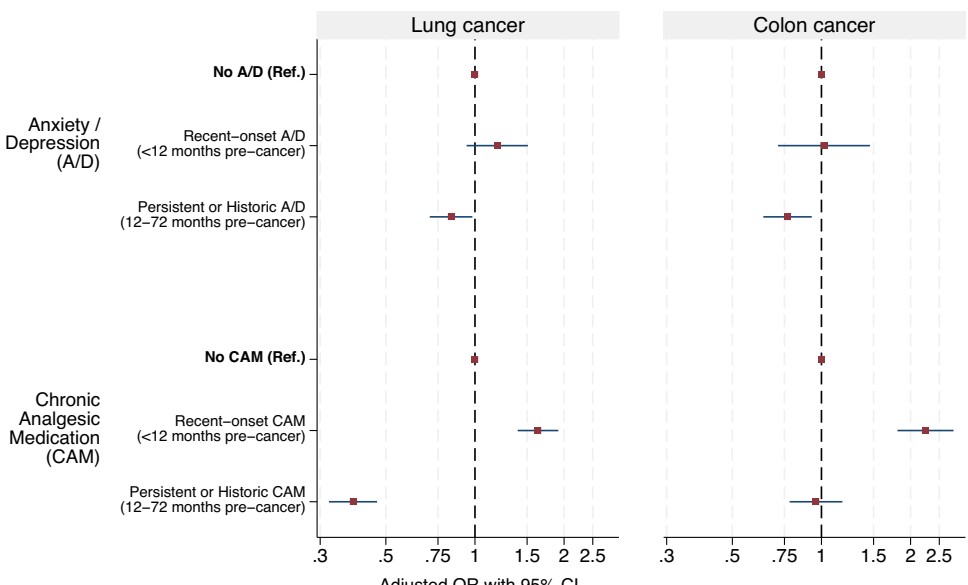

**Fig. 2 | Adjusted Odds Ratios (OR) with 95% Confidence Intervals (CI) of advanced versus early stage cancer diagnosis, by duration of Anxiety/ Depression and Chronic Analgesic Medication use.** Adjusted OR and 95% CI obtained from logistic regression models of i) lung cancer patients (n = 6112) and ii) colon cancer patients (n = 3626). Models adjusted for Anxiety/Depression and Chronic Analgesic Medication use, plus covariates age, sex, IMD income quintile, number of GP consultations, site-specific and general cancer symptoms, number of physical conditions.

with reduced odds of advanced stage colorectal cancer in another. Results also vary among studies using summary measures of pre-existing comorbidity, with some reporting increased comorbidity burden being associated with early stage colorectal cancer[16,17], while others found no evidence of an association[18]. Mixed findings can at least partially be due to differences in how comorbidities are defined, as well as inconsistency regarding timeframes in which information on comorbidity is captured[6]. Our study investigates associations with stage at diagnosis according to the duration and timing of prevalent pre-existing physical and mental health conditions.

Recent-onset Chronic Analgesic Medication use was associated with increased odds of an advanced stage cancer diagnosis. We are unable to infer from our data the length of time patients had an advanced stage of disease prior to their diagnosis – whether the cancers were rapidly progressing or underlying in an advanced stage over a longer period is not known. The mechanisms that can help explain associations between the presence of pre-existing comorbidity and advanced cancer stage are most relevant to rapidly progressing cancers. Acknowledging this, we offer some tentative explanations for how recent-onset conditions may prolong time to diagnosis of cancer, thereby increasing risk of stage progression and diagnosis at a more advanced stage:

i) Recent-onset pain or Anxiety/Depression is triggered by symptoms of underlying as-yet-undiagnosed advanced stage cancer (reverse causality) and treatment of the pain or Anxiety/Depression in parallel to the diagnostic process results in cancer diagnosis occurring later; this is chiefly a manifestation of disease (biological) factors.

ii) Patients are distracted from reporting additional symptoms to their healthcare provider, either because they are primarily focused on dealing with their recent-onset condition (competing demands, influencing help-seeking for cancer symptoms) or because they fear that they may be perceived as hypochondriac[2]. Alternatively, the patient may not report potential cancer symptoms to their healthcare provider as they don't see these symptoms as additional to their pre-existing condition, i.e. they do not recognise the seriousness of symptoms and/ or are unaware of which symptoms relate to which disease[19]. This is chiefly a manifestation of patient (behavioural) factors.

iii) Healthcare providers are focused either on managing complexities of the recent-onset condition, thereby possibly being distracted from investigating additional symptoms (competing demands, influencing the diagnostic interval), or they attribute cancer symptoms erroneously to the recent-onset condition (alternative explanation); this is chiefly a manifestation of healthcare (health system) factors[2].

Our study also provided some evidence that persistent long-standing Chronic Analgesic Medication use prior to cancer diagnosis may be protective against an advanced stage diagnosis. This might be explained by the surveillance mechanism, when increased frequency of consultations with healthcare providers is providing opportunities for early reporting, monitoring and investigation of possible cancer symptoms[2,3].

Approximately half of patients with recent-onset or persistent Chronic Analgesic Medication use had been prescribed opioids. Increased use of opioids prior to cancer diagnosis has been reported in other studies, but published studies examining opioid use and stage at cancer diagnosis are lacking. A study of lung cancer patients in England reported an increase in opioid prescribing in the four months prior to cancer diagnosis, while a Danish study reported increasing use of opioids in the eight months prior to lung cancer diagnosis and four months prior to colon cancer diagnosis[20]. An Australian study of a cohort of cancer survivors noted increased used of opioids in the six months prior to cancer diagnosis among 26% of patients and a constant trajectory of use before diagnosis among 16% of patients[21]. Further research into type and site of pain, time with pain, progression of pain and escalation of pain medication before cancer diagnosis may help disentangle which types of pain are more likely due to the undiagnosed cancer and which are due to other conditions, which may in turn provide opportunities for earlier cancer diagnosis. For example, symptoms of chest pain or back pain prior to the diagnosis of cancer can increase the likelihood of an advanced stage cancer diagnosis[22]. In our study, patients with recent-onset Chronic Analgesic Medication use were more likely to be diagnosed with advanced stage cancer than those without Chronic Analgesic Medication use. By contrast, patients with persistent, longer-term Chronic Analgesic Medication use were less likely to be diagnosed at an advanced stage, suggesting that presenting with pain earlier, allowing for investigation of the pain and other symptoms, may have helped with earlier diagnosis. The CANBACK study, led by researchers at

Oxford University, uses data from patients presenting to primary care with back pain to develop diagnostic models to identify risk of undiagnosed cancer in patients with a new episode of back pain.

We combined Anxiety and Depression in our main analyses, but additional analyses examining these two conditions separately produced unexpected findings. We had anticipated patients with Anxiety would be less likely to be diagnosed with advanced stage cancer (based upon the surveillance mechanism and seeking healthcare more frequently), while those with Depression more likely diagnosed at advanced stage (due to the effect of competing demands or worry to be seen as hypochondriacal). However, our analysis indicated that recent-onset Anxiety was associated with increased odds of advanced stage lung or colon cancer, while historic or persistent Depression was associated with lower odds of advanced stage cancer. Longer-term Depression reducing the odds of advanced stage cancer may be explained by the surveillance effect. The underlying reason behind the association between recent-onset Anxiety and advanced stage cancer diagnosis is less apparent, but may be influenced by reverse causality: symptoms of advanced lung or colon cancer such as dyspnoea or abdominal pain, respectively, may cause anxiety. Given the small percentages of patients in our data with Anxiety and without Depression (and vice versa) (Supplementary Fig. 4), caution is needed when drawing conclusions from these findings. An inability to recognise and/or articulate new cancer-related symptoms to healthcare providers may be influential in associations between pre-existing mental health conditions and timeliness of cancer diagnosis. For example, one vignette study found that patients with Anxiety or Depression were more likely to attribute symptoms of colorectal cancer to mental health conditions rather than to cancer[23].

We acknowledge some limitations with our study. Firstly, approximately 10% of lung cancer patients and 15% of colon cancer patients had missing information on stage at diagnosis, our outcome of interest. We excluded these patients from our multivariable analyses given the small percentage of missingness, and because other methods of handling missing data, such as multiple imputation, have been shown to be less efficient when, as in our analysis, missing data relate to the outcome variable[24]. Second, the presence of mental health conditions may be underreported in administrative health data. However, to mitigate this, we captured relevant information on mental health conditions from both primary care consultation and prescription data. Third, we have adjusted for multimorbidity by using a simple count of conditions. In theory it may be more attractive to consider adjusting for specific combinations of conditions, but in reality this is complex. For example, for 19 specific conditions conventionally included in the Charlson Comorbidity Index[25], there are 171 pairwise combinations, and over half a million possible combinations overall. Adjusting for a small number of specific conditions, selected on the basis they may influence the diagnostic process of cancer, may be an alternative solution when adjusting for multimorbidity in future work. Using colon cancer diagnosis as an example, this would include conditions such as history of Cardiovascular Disease, which has been shown to reduce the likelihood of receiving colonoscopy for investigation of cancer symptoms[26], or Inflammatory Bowel Disease, a condition providing an 'alternative explanation' for colon cancer symptoms, which has been shown to delay cancer diagnosis[27].

To our knowledge, our study is the first to examine the association of specific physical and mental health conditions with stage at cancer diagnosis while explicitly examining the likely influence of the timing of first presentation with comorbidities and their duration. We found that the direction of associations differed according to whether patients had a chronically active disease over longer periods of time or whether the diagnosis of the conditions was recent. These findings are relevant to the management of patients presenting to primary care with new pain, particularly where potential cancer symptoms are also present. Further research into type and severity of recent onset pain, in combination with other symptoms, may help identify groups of patients experiencing pain who are at greater risk of undiagnosed cancer, and of prolonged intervals and advanced stage at diagnosis. Furthermore, it would be informative to conduct further research to understand how timing and duration of these physical and mental health morbidities may influence stage at diagnosis of other cancers.

## Methods

### Ethical approval

The research conducted in this study complies with all relevant ethical regulations. The study was approved by the MHRA (UK) Independent Scientific Advisory Committee (ISAC, ref:18_299R), under Section 251 of the NHS Social Care Act 2006. Our study uses anonymised, secondary data provided by patients and collected by the National Health Service (NHS) as part of their care and support. The NHS has a National Data Opt-Out service, which enables people to decide whether their anonymised confidential patient information can be used for research and planning.

### Study sample and data sources

We used data on incident lung or colon cancer (diagnosed 2012 to 2018) from the England National Cancer Registry, linked with primary care (Clinical Practice Research Datalink, CPRD) data. The CPRD Gold dataset provides primary care patient data from a sample of approximately 6.9% of the UK population, representative of the population in terms of age, sex and ethnicity[28].

Registry data provided information on patient characteristics (age at diagnosis, sex and deprivation quintile of the Income domain of the Indices for Multiple Deprivation[29], an area-based measure of deprivation, based on the Lower layer Super Output Area (LSOA) – a geographic area of mean population 1500 people - the patient resided in at the time of diagnosis) and diagnostic information (cancer site, date of diagnosis and stage of diagnosis). Primary care records (including prescription and blood test data) were used to derive information on pre-existing physical and mental health comorbidities, (cancer-site specific and general) symptomatic presentations and frequency and timing of GP consultations. We used a look-back time window of up to 72 months prior to cancer diagnosis to obtain information on the presence of pre-existing conditions, which has been shown to be a reasonable time window to capture the majority of information available[30].

All data used in our analyses are anonymised secondary data. Our data include a variable indicating patient sex (male/female). We provide aggregate data on the sex distribution of our patient cohorts and adjust for patient sex in our statistical analyses.

### Study variables

Our outcome of interest, stage at diagnosis, was defined according to the rules of the UICC TNM classification system[31]. We categorised stage as early (summary stages I or II) or advanced (summary stages III or IV).

We derived information on thirty-five pre-existing physical and mental health comorbidities from GP consultation records and prescription data using 'Readcode' and 'Prodcode' coding definitions from the Cambridge Multimorbidity Score (CMS)[32] (Supplementary Table 6). 'Readcode' is a data field in CPRD to record patient findings and procedures, based on codes used within the National Health Service, while 'Prodcode' is a CPRD data field representing product codes describing drug and appliance prescriptions. We further defined the thirty-five conditions according to when they were recorded in primary care. In our descriptive analysis we used the following categories: 'Recent-onset' (only recorded <12 months prior to cancer diagnosis), 'Historic' (only recorded 12-72 months pre-cancer), and 'Persistent' (recorded <12 months pre-cancer and also 12-72 months pre-cancer). We then chose to focus on the most prevalent physical condition and - after also considering serious mental health conditions (e.g. Schizophrenia) - the most prevalent mental health condition as explanatory variables in our analyses.

**Table 2 | Distribution of clinical and healthcare characteristics of patients diagnosed with lung cancer between 2012 and 2018, by stage at diagnosis**

| Lung cancer patients | Stage I n | Stage I % | Stage II n | Stage II % | Stage III n | Stage III % | Stage IV n | Stage IV % | Early (Stages I or II) % | Advanced (Stages III or IV) % | Missing n | Missing % | p-value[a] | All patients N | All patients % |
|---|---|---|---|---|---|---|---|---|---|---|---|---|---|---|---|
| **Number of physical morbidities** | | | | | | | | | | | | | <0.01 | | |
| 0 | 238 | 24.8 | 111 | 23.0 | 418 | 32.0 | 1,231 | 36.6 | 24.2 | 35.3 | 235 | 32.8 | | 2233 | 32.7 |
| 1 | 263 | 27.4 | 140 | 29.0 | 383 | 29.3 | 1,012 | 30.1 | 28.0 | 29.9 | 219 | 30.6 | | 2017 | 29.5 |
| 2 | 199 | 20.8 | 114 | 23.7 | 266 | 20.4 | 641 | 19.0 | 21.7 | 19.4 | 150 | 20.9 | | 1370 | 20.1 |
| 3+ | 259 | 27.0 | 117 | 24.3 | 239 | 18.3 | 481 | 14.3 | 26.1 | 15.4 | 112 | 15.6 | | 1208 | 17.7 |
| **Number of mental health morbidities** | | | | | | | | | | | | | <0.01 | | |
| 0 | 641 | 66.8 | 344 | 71.4 | 918 | 70.3 | 2,523 | 75.0 | 68.4 | 73.7 | 520 | 72.6 | | 4,946 | 72.4 |
| 1 | 169 | 17.6 | 85 | 17.6 | 258 | 19.8 | 644 | 19.1 | 17.6 | 19.3 | 155 | 21.6 | | 1,311 | 19.2 |
| 2+ | 149 | 15.5 | 53 | 11.0 | 130 | 10.0 | 198 | 5.9 | 14.0 | 7.0 | 41 | 5.7 | | 571 | 8.4 |
| **Number of GP consultations 30 days to 2 years prior to cancer diagnosis** | | | | | | | | | | | | | <0.01 | | |
| 0-9 | 65 | 6.8 | 34 | 7.1 | 139 | 10.6 | 436 | 13.0 | 6.9 | 12.3 | 55 | 7.7 | | 729 | 10.7 |
| 10-19 | 318 | 33.2 | 203 | 42.1 | 602 | 46.1 | 1468 | 43.6 | 36.2 | 44.3 | 249 | 34.8 | | 2840 | 41.6 |
| 20+ | 576 | 60.1 | 245 | 50.8 | 565 | 43.3 | 1461 | 43.4 | 57.0 | 43.4 | 412 | 57.5 | | 3259 | 47.7 |
| **Haemoptysis** | | | | | | | | | | | | | 0.11 | | |
| No | 930 | 97.0 | 461 | 95.6 | 1260 | 96.5 | 3286 | 97.7 | 96.5 | 97.3 | 691 | 96.5 | | 6628 | 97.1 |
| Yes | 29 | 3.0 | 21 | 4.4 | 46 | 3.5 | 79 | 2.3 | 3.5 | 2.7 | 25 | 3.5 | | 200 | 2.9 |
| **Cough or Dyspnoea** | | | | | | | | | | | | | <0.01 | | |
| No | 357 | 37.2 | 189 | 39.2 | 568 | 43.5 | 1665 | 49.5 | 37.9 | 47.8 | 312 | 43.6 | | 3091 | 45.3 |
| Yes | 602 | 62.8 | 293 | 60.8 | 738 | 56.5 | 1700 | 50.5 | 62.1 | 52.2 | 404 | 56.4 | | 3737 | 54.7 |
| **Fatigue or Weight loss** | | | | | | | | | | | | | 0.11 | | |
| No | 846 | 88.2 | 422 | 87.6 | 1156 | 88.5 | 3024 | 89.9 | 88.0 | 89.5 | 641 | 89.5 | | 6089 | 89.2 |
| Yes | 113 | 11.8 | 60 | 12.4 | 150 | 11.5 | 341 | 10.1 | 12.0 | 10.5 | 75 | 10.5 | | 739 | 10.8 |
| **Anaemia (blood test confirmed)** | | | | | | | | | | | | | 0.06 | | |
| No | 715 | 74.6 | 373 | 77.4 | 1007 | 77.1 | 2630 | 78.2 | 75.5 | 77.9 | 452 | 63.1 | | 5177 | 75.8 |
| Yes | 244 | 25.4 | 109 | 22.6 | 299 | 22.9 | 735 | 21.8 | 24.5 | 22.1 | 264 | 36.9 | | 1651 | 24.2 |
| **Total by stage** | 959 | 14 | 482 | 7.1 | 1306 | 19.1 | 3365 | 49.3 | 21.1 | 68.4 | 716 | 10.5 | | 6828 | 100.0 |

[a]Pearson's Chi-squared test (two-sided), limiting to those with complete stage information, was used to test for an association between each listed characteristic and stage at diagnosis (p = <0.05).

The CMS condition of 'Painful Conditions' was the most prevalent physical condition, which was indicated by ≥4 prescription-only analgesics or ≥4 specified anti-epileptics (with no epilepsy Readcode ever recorded) prescribed during a 12-month period. Recent onset Painful Conditions indicated receipt of 4 such prescriptions during the 12 months before diagnosis, but none prior to this. To increase clarity on the meaning of this condition, we renamed Painful Conditions as 'Chronic Analgesic Medication'.

Anxiety or Depression - the most prevalent mental health conditions - were grouped together under the CMS definitions, given the known correlation between the two conditions[33], representing patients with a Readcode for either Depression or Anxiety, or patients receiving ≥4 anxiolytic/hypnotic or ≥4 anti-depressant prescriptions in a 12-month period.

We quantified the frequency/timing of GP consultations according to the number of consultations occurring during the 24 months prior to cancer diagnosis, assuming consultations in the more distant past were less likely to be associated with the underlying cancer. We excluded consultations occurring 30 days or less before diagnosis, assuming them to be directly related to the cancer diagnosis. We defined cancer site-specific and general cancer symptoms based upon NICE guidelines[34] and previous literature[22]. Using information relating to these symptoms in the 'Readcode' and 'Prodcode' CPRD data fields, we then identified patients with these symptoms pre-cancer from GP consultations occurring between 1-6 months before cancer diagnosis. Site-specific symptoms included haemoptysis, cough or dyspnoea for lung cancer, and change in bowel habit, rectal bleeding, abdominal pain/bloating, constipation or diarrhoea for colon cancer. General

symptoms of lung or colon cancer were fatigue or weight loss. Anaemia was derived from haemoglobin blood test results and NICE guidelines[35] for defining anaemia. Colon cancer symptoms were categorised as "red-flag symptoms" (change in bowel habit or rectal bleeding) and "only non-red-flag symptoms" (abdominal pain/bloating, constipation, diarrhoea, in the absence of red-flag symptoms). We calculated the total number of physical conditions, as a measure of comorbidity burden. We illustrated our hypothesised relationships between Chronic Analgesic Medication use or Anxiety/Depression, covariates and stage at cancer diagnosis in a Directed Acyclic Graph (Supplementary Fig. 5), using DAGitty software[36].

### Patient characteristics

Using ICD-10 codes in the registry data we identified 13,444 patients diagnosed with lung cancer (ICD-10 code 'C34') and 8345 patients diagnosed with colon cancer (ICD-10 code 'C18') between 2012-2018. We excluded patients i) without a record of a GP consultation in the 12 months before cancer diagnosis (to ensure all patients included in our analysis had an equal baseline for capturing information on pre-existing conditions), ii) not registered at an Up To Standard CPRD practice for at least 12 months prior to diagnosis from our study sample and iii) with missing patient sociodemographic information. Following these exclusions, 6828 lung cancer patients and 4194 colon cancer patients were included in the study (details in Supplementary Fig. 1).

Over two-thirds (68%) of the 6828 patients diagnosed with lung cancer had an advanced stage lung cancer diagnosis, while a fifth (21%) had an early-stage diagnosis, and stage was missing for approximately 10% of patients (Table 2, Supplementary Table 7). All covariates, except

**Table 3 | Distribution of clinical and healthcare characteristics of patients diagnosed with colon cancer between 2012-2018, by stage at diagnosis**

| Colon cancer patients | Stage I n | Stage I % | Stage II n | Stage II % | Stage III n | Stage III % | Stage IV n | Stage IV % | Early (Stages I or II) % | Advanced (Stages III or IV) % | Missing n | Missing % | p-value[c] | All patients N | All patients % |
|---|---|---|---|---|---|---|---|---|---|---|---|---|---|---|---|
| **Number of physical morbidities** | | | | | | | | | | | | | <0.01 | | |
| 0 | 178 | 35.1 | 422 | 38.9 | 376 | 39.2 | 460 | 42.9 | 37.7 | 41.1 | 225 | 39.6 | | 1661 | 39.6 |
| 1 | 153 | 30.2 | 322 | 29.7 | 295 | 30.7 | 340 | 31.7 | 29.8 | 31.2 | 164 | 28.9 | | 1274 | 30.4 |
| 2 | 74 | 14.6 | 190 | 17.5 | 151 | 15.7 | 170 | 15.8 | 16.6 | 15.8 | 80 | 14.1 | | 665 | 15.9 |
| 3+ | 102 | 20.1 | 152 | 14.0 | 138 | 14.4 | 103 | 9.6 | 15.9 | 11.9 | 99 | 17.4 | | 594 | 14.2 |
| **Number of mental health morbidities** | | | | | | | | | | | | | 0.02 | | |
| 0 | 398 | 78.5 | 812 | 74.8 | 738 | 76.9 | 864 | 80.5 | 76.0 | 78.8 | 446 | 78.5 | | 3258 | 77.7 |
| 1 | 58 | 11.4 | 165 | 15.2 | 136 | 14.2 | 143 | 13.3 | 14.0 | 13.7 | 82 | 14.4 | | 584 | 13.9 |
| 2+ | 51 | 10.1 | 109 | 10.0 | 86 | 9.0 | 66 | 6.2 | 10.0 | 7.5 | 40 | 7.0 | | 352 | 8.4 |
| **Number of GP consultations 30 days to 2 years prior to cancer diagnosis** | | | | | | | | | | | | | 0.13 | | |
| 0-9 | 62 | 12.2 | 150 | 13.8 | 140 | 14.6 | 178 | 16.6 | 13.3 | 15.6 | 73 | 12.9 | | 603 | 14.4 |
| 10-19 | 234 | 46.2 | 532 | 49.0 | 465 | 48.4 | 499 | 46.5 | 48.1 | 47.4 | 230 | 40.5 | | 1960 | 46.7 |
| 20+ | 211 | 41.6 | 404 | 37.2 | 355 | 37.0 | 396 | 36.9 | 38.6 | 36.9 | 265 | 46.7 | | 1631 | 38.9 |
| **Red-flag[a] symptoms** | | | | | | | | | | | | | 0.04 | | |
| No | 424 | 83.6 | 971 | 89.4 | 850 | 88.5 | 975 | 90.9 | 87.6 | 89.8 | 508 | 89.4 | | 3728 | 88.9 |
| Yes | 83 | 16.4 | 115 | 10.6 | 110 | 11.5 | 98 | 9.1 | 12.4 | 10.2 | 60 | 10.6 | | 466 | 11.1 |
| **Only non-red-flag[b] symptoms** | | | | | | | | | | | | | <0.01 | | |
| No | 389 | 76.7 | 711 | 65.5 | 633 | 65.9 | 661 | 61.6 | 69.1 | 63.6 | 365 | 64.3 | | 2759 | 65.8 |
| Yes | 118 | 23.3 | 375 | 34.5 | 327 | 34.1 | 412 | 38.4 | 30.9 | 36.4 | 203 | 35.7 | | 1435 | 34.2 |
| **Anaemia (in absence of red-flag symptoms)** | | | | | | | | | | | | | 0.81 | | |
| No | 368 | 72.6 | 665 | 61.2 | 622 | 64.8 | 704 | 65.6 | 64.8 | 65.2 | 338 | 59.5 | | 2697 | 64.3 |
| Yes | 139 | 27.4 | 421 | 38.8 | 338 | 35.2 | 369 | 34.4 | 35.2 | 34.8 | 230 | 40.5 | | 1497 | 35.7 |
| **Total by stage** | 507 | 12.1 | 1,086 | 25.9 | 960 | 22.9 | 1073 | 25.6 | 38.0 | 48.5 | 568 | 13.5 | | 4194 | 100.0 |

[a] Red-flag symptoms: Change in bowel habits and rectal bleeding.
[b] Non-red flag symptoms: abdominal pain/bloating, constipation or diarrhoea.
[c] Pearson's Chi-squared test (two-sided), limiting to those with complete stage information, was used to test for an association between each listed characteristic and stage at diagnosis ($p = <0.05$).

deprivation quintile, and having records of Haemoptysis, Fatigue or Weight Loss, or Anaemia, were associated with stage at diagnosis ($p < 0.05$).

Almost half (49%) of the 4194 patients diagnosed with colon cancer had an advanced stage colon cancer diagnosis, 38% were diagnosed at an early stage, and 13% of patients had missing stage (Table 3, Supplementary Table 8). The variables associated with stage ($p < 0.05$) were the number of physical morbidities, number of mental health morbidities, having red-flag symptoms or only having non-red flag symptoms.

**Statistics and reproducibility**

We summarised the distribution of patient, clinical and healthcare characteristics and used cross tabulations to summarise the distributions of characteristics by stage at diagnosis. Pearson's Chi-squared test, limiting to those with complete stage information, was used to assess whether observed differences in stage distributions by patient characteristics were consistent with chance. We calculated the prevalence of 35 different pre-existing comorbidities, based on the percentage of patients with each condition, and used this information to summarise the most prevalent pre-existing conditions by cancer site. We then categorised prevalence according to when the conditions were recorded in primary care. For subsequent analysis we focused on Chronic Analgesic Medication use and Anxiety/Depression as these

were the most prevalent physical and mental health conditions. We summarised the type of drugs prescribed to patients with Chronic Analgesic Medication use, as this was derived solely from prescription data.

We used multivariable logistic regression modelling to estimate associations between Anxiety/Depression and Chronic Analgesic Medication use with stage at cancer diagnosis. In the models these conditions were categorised as either 'Not present', 'Recent-onset' or 'Persistent/Historic'. We adjusted the model for sociodemographic variables (age, sex, deprivation quintile), GP consultations, total number of physical comorbidities and presence of site-specific or general cancer symptoms. We ran a separate sensitivity analysis using the same multivariable logistic regression models but with stage redefined as early (stages I-III) and advanced (stage IV), to compare with our main analysis results.

We also ran separate analyses to test for interactions between the pre-existing comorbidities and other covariates. In these multivariable logistic regression models we tested for interactions between i) Anxiety/Depression and age or sex, ii) Chronic Analgesic Medication use and age or sex, iii) Anxiety/Depression and Chronic Analgesic Medication use.

Lastly, we performed a sensitivity analysis, re-running the models of the main analysis but modelling Anxiety and Depression separately rather than as a combined condition, to see if the two conditions had similar associations with stage at diagnosis.

Data management was conducted using MySQL Workbench Version 8[37] and all analyses conducted using Stata 18[38].

## Reporting summary

Further information on research design is available in the Nature Portfolio Reporting Summary linked to this article.

## Data availability

The data used in this study were obtained under licence from CPRD. Access to the data can be requested via application to CPRD. Applicants must submit a research study protocol, requesting the specific data required, via the CPRD electronic Research Application Portal. This portal, along with guidance on submitting applications, is accessed via the CPRD website (https://www.cprd.com).

## Code availability

The code used in this analysis are available via the GitHub repository: https://github.com/cfiles-hf2025/analysis_code_files_ad_cam_stage_diagnosis.

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

## Acknowledgements

This study is based on anonymised data from the Clinical Practice Research Datalink (CPRD), obtained under licence from the UK Medicines and Healthcare products Regulatory Agency. The data is provided by patients and collected by the NHS as part of their care and support. The interpretation and conclusions contained in this study are those of the author/s alone. This study was funded by the NIHR Policy Research Programme (reference PR-PRU-NIHR206132)(GL, MEB); the Cancer Research UK—Early Detection and Diagnosis Committee (grant number EDDCPJT\100018)(CR, GL, HF); and the International Alliance for Cancer Early Detection (ACED), a partnership between Cancer Research UK (CRUK; reference: C18081/A31373, RREDD-EHR project)(GL, CR, MR, HF), Canary Center at Stanford University, the University of Cambridge, Oregon Health and Science University Knight Cancer Institute, University College London, and the University of Manchester. MEB is supported by a Cancer Research UK ACED Pathway Award Fellowship (reference: EDDAPA-2022/100002). The views expressed are those of the author(s) and not necessarily those of the NIHR or the Department of Health and Social Care. Symptoms were defined using libraries of Read codes developed by Professor Willie Hamilton and Dr Sarah Price at Exeter University, with additional codes added by colleagues Professor Georgios Lyratzopoulos, Dr Cristina Renzi, Dr Becky White, Dr Matthew Barclay, and Dr Meena Rafiq at UCL.

## Author contributions

Conceptualisation: CR, HF; Supervision: CR, GL; Data management and analysis: HF; Drafting manuscript: HF; Reviewing and editing manuscript: CR, GL; Reviewing and editing final draft of manuscript: MR, MEB, GAA

## Competing interests

The authors declare no competing interests.
