## [Transparent Peer Review file · Nature Communications]

Cancer stage at diagnosis by duration of pre-existing Chronic Analgesic Medication and Anxiety or Depression

Corresponding Author: Dr Helen Fowler

Version 0:

Reviewer comments:

Reviewer #1

(Remarks to the Author)
NCOMMS-24-79057

The authors use data from the England National Cancer Registry and linked primary care data to understand whether patients with a variety of pre-existing conditions including anxiety/depression are at higher risk of being diagnosed with advanced stage lung and colon cancer. In summary, they conclude that the odds of an advanced stage at cancer diagnosis are higher in patients with recent-onset medical conditions vs. those without.

My concerns with the paper fall into two general categories.

1. Clinical relevance:

- a. The authors cite a fairly large body of literature that has looked at the association between various co-morbidities and cancer stage at diagnosis, and the novelty of this paper seems to be their ability to distinguish between co-morbidities with different timing (i.e., recent-onset, historic, and persistent). However, I do not think this represents an important distinction and may unnecessarily complicate the analysis; for example, the difference between a patient having had atrial fibrillation for 12 or 24 months should not be relevant to their cancer stage at diagnosis. Along those lines, the definitions of historic, persistent, and recent-onset seem a bit contrived, and using only “acute” and “chronic” may be simpler as well as more consistent with how clinicians typically consider co-morbidities.
- b. Why is the analysis limited to patients with lung and colon cancer? Including all malignancies would presumably make for a more robust study.
- c. “Painful condition” is a very vague term and does not represent a distinct medical diagnosis, so I ultimately feel it needs to be removed from the analysis. Alternately, I believe the authors have the data to characterize a more precise and meaningful diagnosis such as “chronic opioid use.”
- d. It is not unreasonable to collapse stages I/II and III/IV for the purposes of the logistic regression model, but the authors should present/discuss the complete data (i.e., separately reporting stage I, II, III, and IV).

2. Statistical methods:

- a. The authors appear to have run separate logistic regressions for the outcome of recent-onset, historic, and then persistent anxiety/depression; the model for painful conditions was similarly split, then they also built a model with both conditions combined. This approach does not have compelling clinical rationale, and increases the chance of finding a spurious association due to over-testing and smaller sample sizes (for instance, only 2.8% of patients have “historic painful conditions”).
- b. The prevalence of co-morbidities other than anxiety/depression and “painful conditions” are reported in supplementary Table 2 but do not appear to be included in any of the models?

A few other comments:

- It is not clear precisely how symptoms are recorded, and the terms “readcode”/“medcode” need to be defined. I imagine diagnoses are entered in the patient’s chart in an extractable

Reviewer #2

(Remarks to the Author)

This is a large cohort study from the UK looking at symptoms before a cancer diagnosis. It is part of studies in constant search for indications that a cancer disease is ongoing in a person before the diagnosis is established. Screening is another form of detecting a cancer disease at an early stage with or without symptoms or signs which may indicate a cancer disease. In this study the authors looked at different durations of symptoms making these symptoms chronic as experienced by the person having these symptoms

In the study the authors included lung and colon cancer patients in a register-based fashion over a seven-year period (2012 to 2018). Patients were identified by a linkage between a cancer register and general practitioner register. The authors observed that pain conditions (PC) and anxiety/depression (AD) were the most prevalent conditions diagnosed before a cancer diagnosis and then divided the exposure of these conditions into a recent, historic or persistent exposure time-period.

Patients with a recent onset of such PC were at an increased risk for advanced stage colon or lung cancer. On the other hand, having persistent PC reduced the risk for advanced stage lung cancer but not for advanced stage colon cancer. This pattern was also observed in patients suffering from AD conditions.

The authors suggest that the observations indicate that persistent conditions offer more opportunities for an early diagnosis of a cancer disease. More recent diagnosed conditions may lead to a delay in a cancer diagnosis.

My comments:

The background literature review to some degree miss the variations in the field, different approaches and knowledge obtained in studies not directly comparable to the study under review but giving insight into the problem of symptoms, behavior or morbidities existing at the time of a cancer diagnosis. Several studies from the Nordic countries using population-based and often nationwide registries as data sources have looked at the same pattern or existence of symptoms or diseases at time of the cancer diagnosis. One study examined blood sample activity up to eight years before a hematology cancer diagnosis another the existence of morbidity and prescribed medications at time of the cancer diagnosis illustrating different aspects of the current study.

Seen from a clinical point of view recent observed conditions in a patient without a direct explanation would lead to further examinations to exclude malignancy. Some but not all cancers are identified due to pain issues but no matter the treatment of pain, patients who return to the GP with pain issues following the prescription of pain killers would enter a program for identifying a cancer disease. This would follow a clinical examination by the GP, that probably would point to an anatomical area of interest. Whether the disease is local or advanced is difficult to clearly define at this point in time in the disease trajectory. Pain is probably the most common symptom that any human being will experience through a lifetime, and it is not surprising that this is one of the most common symptoms observed before a cancer diagnosis, no matter the stage. In addition, anxiety and depression are two common conditions, which in industrialized countries probably is prevalent in around 10 % of the population. Aside from 'normal' conditions, e.g., pregnancy, infant childcare, and age depending reductions in functions, these two conditions AD and PC is therefore also the main reason for contacts to the GP and it comes as no surprise that if a person has a chronic PC then the GP is not that fast in activating cancer diagnostic examinations/pathways as the PC may have no relationship with the two cancers under study. The authors do not fully acknowledge, that the PC and/or AD may be symptoms existing isolated from the lung or colon cancer and therefore no indication of these two cancers. The 'trajectory thinking' is too much in focus of the a priori ideas about disease patterns and disease etiology/trajectory.

The exclusion of the 12 months prior to the cancer diagnosis also 'break' the connection or association between the two symptoms chosen and the cancers investigated, and I suggest conducting analysis including this time-period for comparison with the chosen strategy. This may provide new results of interest for understanding the pattern of symptoms before lung and colon cancer. The activity and complains of various symptoms during this time period is probably quite intensive or more events/contacts to the GP will take place and thereby provide more insight into the mechanisms underlying the GP 'behavioral' (clinical tests/referrals/examinations) aspects of this study.

One may also speculate if a cluster analysis would be of interest as both lung and colon cancer patients may have other symptoms, which are associated with the AD and PC, which may provide other ways/aspects into the analytical understanding of the road from symptoms to a cancer diagnosis.

Reviewer #3

(Remarks to the Author)

This paper presents highly interesting novel findings based on rigorous and sound scientific methodology and an excellent and unselected datasample of the population. It is well-written and to the point. Results are important for advancing early diagnostics in cancer to ensure better treatment, survival and quality of life outcomes. The results expands current knowledge in that they add the timing of/duration of the medical conditions that they examine in connection with stage at diagnosis. This makes it clinically highly relevant but of course being based on a retrospective population may warrant further replication but also more detailed analyses to narrow down the "red flag" conditions. The author group provide detailed analyses and identify subgroup of patients i.e. on opioid treatment as specific subgroups. To my mind this paper presents novel data warranting publication.

Below I specify comments to enhance the clarity of the study for the reader and add to the interpretation of results.

Background:

In the review of the evidence, one may consider if magnitude and direction of associations may also differ by symptoms - as qualitative evidence points to the failure of some patients with chronic disorders which may lead them to see their doctor regularly, to recognize or articulate new symptoms as 'their body is making a lot of noise'. This points to health literacy as a driver in differences in stage at diagnosis

The rationale behind choice of lung cancer and colon cancer is not quite clear - should be founded in scientifically sound reasons and not convenience, as I am sure it also is. Please add this information to the background

Materials and methods

The study period is only up to 2018 and thus not as up-to-date as would be optimal, especially with a growing focus on early diagnostics an updated dataset would have been preferred. In nature, registry based data will not be completely up to date, but this study period ends 6 years ago

For the reader, it may be helpful to add that the deprivation indicator is area-based

The outcome is dichotomized but I would like a rationale for including stage III in the advanced stage group - could consider IIIb-IV for lung and IV for colon cancer?

Does the chosen dichotomization align with the treatment protocols in UK during the study period?

A large number of covariates were adjusted for in the multivariate regression models, and I assume these variables are all seen as confounders.

I.e. a lot of the chronic disorders may include symptoms of pain - and thus the associations between painful conditions and number of morbidities or number of consultations may be complex - and some may act on mediators on same associations? Please clarify by presenting a directed acyclic graph to justify selection of confounders - and refrain from adjusting for any mediators.

Results

When reading the detailed and informative results, one thing may have been elaborated on - do we know if there was an association between painful conditions and acute presentation of colon cancer - which may often have a high stage (not always) but have a very high mortality. Would it be possible to include information on this in the dataset, to explore this further? Else, please include as a limitation as they usually amount to some 10% of the bowel cancer group.

Discussion

Is balanced, and well-written.

The suggestion of 3 tentative explanations is bold and well argued - I suggest authors consider to include that the patients may not see symptoms as additional to their condition - as they may not know what is a symptom of what disease - this is relevant addition to the second (patient factors association).

Separating depression and anxiety based on prescription medication is not clear-cut but the discussion of results of the subanalyses is cautious. May results on anxiety indicate that symptoms of advanced lung or colon cancer actually may in themselves cause anxiety (dyspnea, stomach pain).

Reviewer #4

(Remarks to the Author)

This observational study examines the association between stage at diagnosis in colon and lung cancer patients and the type and duration of physical and mental health morbidities. These comorbid conditions were categorized as "recent onset" (< 12 months pre-cancer diagnosis), "historic" (12-72 months pre-cancer diagnosis), and "persistent" (< 12 and 12-72 months pre-diagnosis).

The analysis identified the most prevalent conditions to be anxiety/depression and painful conditions; and findings from the multivariable regression analysis showed that for each of these conditions, recent onset was associated with higher odds of being diagnosed with late-stage colon or lung cancer. Conversely, the presence of these conditions were associated with lower odds of being diagnosed with late-stage disease when they were persistent because patients with persistent conditions may have greater healthcare utilization, and therefore greater opportunities to be diagnosed with earlier-stage disease.

Prior studies have shown mixed results between the presence of comorbid conditions and stage at diagnosis; but the innovative feature of this study is in distinguishing between longstanding and recent comorbidities.

The analysis is straight-forward, and the manuscript is well written. However, there are some major and other weaknesses:

Major weakness:

As in most studies, comorbid conditions are dealt with one at a time; however, comorbidities often present in combination, rather than isolation--or as part of multimorbidity. Such combinations are not taken into account in the analysis.

Other weaknesses:

1- The study uses data from the England National Cancer Registry linked with the Clinical Practice Research Datalink, CPRD data. The CPRD data set provides data for a sample of 6.9% of the UK population. This is a fairly small sample of the population. How confident are the authors that they can generalize their findings to the population overall?

2- What do 'readcode' and 'procode' stand for? I am not familiar with these terms.

3- What is the rationale for selecting colon and lung cancer for the analysis? Why were other cancers left out?

4- The authors state that they decided to focus on painful conditions and depression/anxiety because they were the most prevalent conditions; however, based on the data presented in Supplemental Table 2, there are several other conditions that are prevalent, especially in the "historic" and "persistent" categories. The authors should therefore find a better justification for focusing on painful conditions and depression/anxiety.

5- In the multivariable analysis, could there be an issue of collinearity between the number of visits and the categories of "recent", "historic", and "persistent"?

Version 1:

Reviewer comments:

Reviewer #1

(Remarks to the Author)

I found the responses highly responsive to the comments and laud the authors on their efforts.

Reviewer #2

(Remarks to the Author)

I have no further comments to the extensive responses from your team. The paper reads well in the current edition.

Reviewer #3

(Remarks to the Author)

Thank you for the opportunity to review your revised manuscript. You have addressed all my concerns and as far as I can see also the concerns of the other reviewers. I have no further comments

Reviewer #4

(Remarks to the Author)

The authors have satisfactorily addressed the most of the comments by reviewers.

However, to address the comment on multimorbidity by Reviewer 4, the authors have simply included the number of physical and mental comorbidities in the multivariable models. While this is helpful, the insight gained by looking at combinations of comorbid conditions would be far richer and more informative than accounting for the number of chronic conditions.

This may be outside the scope of this study. At a minimum, however, this should be noted in the limitations, and suggested as a future study.

How Reviewer comments were reflected and addressed

Manuscript: *Variation in cancer stage at diagnosis by onset and duration of pre-existing Painful Conditions and Anxiety or Depression: a population-based longitudinal study of lung and colon cancer patients in England*

Thank you for the constructive reviews and for the opportunity to resubmit a revised version of the manuscript. Hereafter, we provide below all comments received by the Reviewers, alongside our response.

Reviewer 1

Comment received	How reflected and addressed
The authors use data from the England National Cancer Registry and linked primary care data to understand whether patients with a variety of pre-existing conditions including anxiety/depression are at higher risk of being diagnosed with advanced stage lung and colon cancer. In summary, they conclude that the odds of an advanced stage at cancer diagnosis are higher in patients with recent-onset medical conditions vs. those without.	Thank you for the apt high-level summary of the method and findings of our paper.

Comment received	How reflected and addressed
My concerns with the paper fall into two general categories Category 1: Clinical relevance 1a. Distinguishing between co-morbidities with different timing (recent-onset, historic, persistent) does not represent an important distinction and may complicate the analysis (e.g. patient with atrial fibrillation for 12-24 months should not be relevant to cancer stage at diagnosis). Using only “acute” and “chronic” may be simpler as well as more consistent with how clinicians consider co-morbidities.	In line with the Reviewer’s suggestion, after initially analysing the data using the three categories, we have collapsed the “historic” and “persistent” groups into one category in the multivariable analysis, to increase clarity and minimise the number of parameters in multivariable models. The updates have been described in the methods section and illustrated in Figure 2. The text in the Methods (Statistical analysis section, second paragraph) (new text denoted in bold font): “We used multivariable logistic regression modelling to estimate associations between Anxiety / Depression and Chronic Analgesic Medication with stage at cancer diagnosis. We classified these conditions as either ‘Recent-onset’ or ‘Persistent / Historic’.”
1b. Why is analysis limited to lung and colon cancer patients – including all malignancies would make for a more robust study.	We focused on colon and lung cancer as they are among the most common cancers, are frequently diagnosed at advanced stage and via emergency presentation and have high mortality. Additionally, investigating less common cancers independently may result in issues with data sparsity in our sample and lack of statistical power. We updated the Introduction section as follows (last paragraph): “In our study we focus on lung and colon cancers. These are common cancers, frequently diagnosed at advanced stage and via emergency presentation, and having high mortality.” Considering all malignancies (together) would make identifying associations with chronic comorbidities harder, as mechanisms influencing associations between specific comorbidities and stage at cancer diagnosis may vary by cancer site. A study in New Zealand illustrated that associations between specific comorbidities and cancer stage at diagnosis may vary by cancer site. As an example, having heart failure or stroke was associated with increased odds of advanced stage breast cancer, but there was no evidence of an association between these conditions and colorectal cancer stage.²
1c. Painful Condition is a vague term and does not represent a distinct medical diagnosis and should be removed from the analysis. Alternatively, consider a more precise and meaningful diagnosis such as “chronic opioid use”.	Thank you for this suggestion. To clarify the meaning of the term Painful Conditions we have now renamed this condition “Chronic Analgesic Medication” in our study. We have added the following text to the Methods (Study variables section, 4th paragraph) (bold fonts denote new text):

Comment received	How reflected and addressed
	“The CMS condition of ‘Painful Conditions’ was the most prevalent physical condition, which was indicated by ≥ 4 prescription-only analgesics or ≥ 4 specified anti-epileptics (with no epilepsy readcode ever recorded) prescribed during a 12-month period. Recent onset Painful Conditions indicated receipt of 4 such prescriptions during the 12 months before diagnosis, but none prior to this. To increase clarity on the meaning of this condition, we renamed Painful Conditions as ‘Chronic Analgesic Medication’.”
1d. Separately report [distribution of] stage I, II, III and IV	Thank you for this comment. We have now updated Table 1 to include columns to show the distribution of stage I, II, III and IV.
Category 2: Statistical methods 2a. Modelling approach (separate models for PC, AD and then one with both conditions combined) does not have a compelling clinical rationale and increases the chance of finding a spurious association due to over-testing and smaller sample sizes (e.g. only 2.8% of patients have “historic” painful conditions).	Thank you for this suggestion. We have now streamlined the reporting of our results. Our main analysis model includes both Chronic Analgesic Medication (previously Painful Conditions) and Anxiety / Depression combined. As per our previous comment, we have also collapsed the categories of ‘persistent’ and ‘historic’ into one category.
2b. The prevalence of co-morbidities other than anxiety / depression and painful conditions are reported in Supplementary Table 2 but do not appear to be included in any of the models?	Our first objective was to quantify the prevalence of 35 conditions among incident lung and colon cancer patients, and our second objective was to examine the association between the most prevalent physical and most prevalent mental health condition and stage at diagnosis, using multivariable logistic regression analysis. Although we initially examined the prevalence of individual conditions, in the multivariable analysis we included an explanatory variable denoting the total number of physical or mental health conditions, to account for the presence of other conditions.
Other comments: It is not clear how symptoms are recorded	We updated the Methods as follows (new text in bold) (Study variables section, 5th paragraph): “We defined cancer site-specific and general cancer symptoms based upon NICE guidelines and previous literature. Using information relating to these symptoms in the ‘readcode’ and ‘prodcode’ CPRD data fields, we then identified patients with these symptoms pre-cancer from GP consultations occurring between 1-6 months before cancer diagnosis.”
The terms “readcode” and “medcode” need to be defined	We defined these terms in the Methods section of the paper (‘Study variables’ section, 2nd paragraph) - added text in bold below: “We derived information on thirty-five pre-existing physical and mental health comorbidities from GP consultation records and prescription data using ‘readcode’ and ‘prodcode’ coding definitions from the Cambridge Multimorbidity Score (CMS) (Supplementary Table 1). ‘Readcode’ is a data field in CPRD to record patient findings and procedures, based on

Comment received	How reflected and addressed
	codes used within the National Health Service, while 'prodcode' is a CPRD data field representing product codes describing drug and appliance prescriptions."

Reviewer 2

Comment received	How reflected and addressed
“This is a large cohort study from the UK looking at symptoms before a cancer diagnosis. It is part of studies in constant search for indications that a cancer disease is ongoing in a person before the diagnosis is established. Screening is another form of detecting a cancer disease at an early stage with or without symptoms or signs which may indicate a cancer disease. In this stud the authors looked at different durations of symptoms making these symptoms chronic as experienced by the person having these symptoms In the study the authors included lung and colon cancer patients in a register-based fashion over a seven-year period (2012 to 2018). Patients were identified by a linkage between a cancer register and general practitioner register. The authors observed that pain conditions (PC) and anxiety/depression (AD) were the most prevalent conditions diagnosed before a cancer diagnosis and then divided the exposure of these conditions into a recent, historic or persistent exposure time-period. Patients with a recent onset of such PC were at an increased risk for advanced stage colon or lung cancer. On the other hand, having persistent PC reduced the risk for advanced stage lung cancer but not for advanced stage colon cancer. This pattern was also observed in patients suffering from AD conditions. The authors suggest that the observations indicate that persistent conditions offer more opportunities for an early diagnosis of a cancer disease. More recent diagnosed conditions may lead to a delay in a cancer diagnosis.”	Thank you very much for engaging with our study and its methods and findings and the useful summary.

Comment received	How reflected and addressed
[1] Background literature misses some variations in the field – studies not directly comparable but giving insight into the problem of symptoms, behaviour or morbidities existing at the time of a cancer diagnosis – e.g. one Nordic study of blood sample activity up to eight years before a hematology cancer diagnosis, another the existence of morbidity and prescribed medications at time of cancer diagnosis.	Thank you for this comment. We have added text to the Introduction (second paragraph) to develop the statement on interpretation of cancer symptoms in the presence of pre-existing conditions (added text in bold): “Additionally, the impact of treatment for pre-existing chronic conditions, for example opioid treatment for chronic pain, may influence the interpretation of cancer symptoms, and ultimately impact timing of cancer diagnosis. Similarly, use of several different prescription medications for the treatment of pre-existing conditions may also complicate the interpretation of cancer symptoms. A population-based Danish study of twenty cancers found that one-third of patients had received prescriptions for 5 or more medications during the year before their cancer diagnosis. The presence of pre-existing conditions may also increase the likelihood of patients presenting to their GP with non-specific symptoms. Routine blood testing to identify the cause of the symptoms may lead to cancer being detected following further investigation of abnormal blood results. A Danish study investigating the probability of a cancer diagnosis among a group of patients undergoing routine blood tests for non-specific serious symptoms found that patients with specific combinations of two abnormal blood tests had up to 62% probability of having cancer. While the impact on diagnostic pathways and cancer stage might vary depending on the timing and duration of chronic conditions, typically studies have not distinguished between long-standing or recent-onset conditions.
[2] The authors do not fully acknowledge that the PC and / or AD may be symptoms existing isolated from the lung or colon cancer and therefore no indication of these two cancers. The ‘trajectory thinking’ is too much in focus of the a priori ideas about disease patterns and disease etiology / trajectory.	Thank you for this comment. We agree that these conditions may not be directly related to the cancer. However, the presence of these conditions may still interfere with the diagnostic process of cancer, for example, by distracting a patient from reporting other symptoms that may be related to cancer to their GP.
[3] The exclusion of the 12 months prior to cancer diagnosis also ‘break’ the connection or association between the two symptoms chosen and the cancers investigated, and I suggest conducting analysis including this time-period for comparison with the chosen strategy. This may provide new results of interest for understanding the pattern of symptoms before lung and colon cancer.	We did include the period of 12 months prior to cancer diagnosis, as also shown in the Methods section, where we describe categorising conditions according to when they were recorded in primary care (‘Study variables’ section, third paragraph): “... we used the following categories: ‘Recent-onset’ (only recorded <12 months prior to cancer diagnosis)...’Persistent’ (recorded <12 months pre-cancer and also 12-72 months pre-cancer)”. The labelling in our Results (e.g. Figures 1 and 2) also illustrates when the conditions were recorded in primary care.

Comment received	How reflected and addressed
	We did find differences in associations with stage at diagnosis by whether the conditions were diagnosed either <12 or >12 months before the cancer diagnosis. Specifically, we found that having recent-onset Chronic Analgesic Medication [Painful Conditions] versus none increased the odds of having advanced stage lung and colon cancer (adjusted Odds Ratio (aOR):1.63; 95% CIs: 1.39, 1.91 and aOR: 2.25; 95%Cis 1.81, 2.79, respectively). Conversely, having Persistent / Historic Chronic Analgesic Medication versus none reduced the odds of advanced stage lung cancer diagnosis (aOR 0.39; 95%CI: 0.32, 0.47) but was not associated with colon cancer stage at diagnosis (aOR: 0.96; 95%CIs: 0.78, 1.18).
[4] Would a cluster analysis be of interest as both lung and colon cancer patients may have other symptoms, which are associated with the AD and PC, which may provide other ways / aspects into the analytical understanding of the road from symptoms to cancer diagnosis.	Thank you for this suggestion. In our study we took a slightly different approach to a cluster analysis. We considered non-localising symptoms (such as fatigue and weight loss) and also localising cancer-site-specific symptoms (such as abdominal pain and constipation for colon cancer and cough or dyspnoea for lung cancer). We agree that it would interesting to also look at a broader range of symptoms as part of a cluster analysis, but think the scope of this work would be better placed as a separate analysis and something we could consider for further research.

Reviewer 3

Comment received	How reflected and addressed
This paper presents highly interesting novel findings based on rigorous and sound scientific methodology and an excellent and unselected data sample of the population. It is well-written and to the point. Results are important for advancing early diagnostics in cancer to ensure better treatment, survival and quality of life outcomes. The results expands current knowledge in that they add the timing of/duration of the medical conditions that they examine in connection with stage at diagnosis. This makes it clinically highly relevant but of course being based on a retrospective population may warrant further replication but also more detailed analyses to narrow down the "red flag" conditions. The author group provide detailed analyses and identify subgroup of patients i.e. on opioid treatment as specific subgroups. To my mind this paper presents novel data warranting publication. Below I specify comments to enhance the clarity of the study for the reader and add to the interpretation of results.	Thank you very much for the very supportive and constructive assessment and your subsequent useful suggestions.
[1] In the review of the evidence, one may consider if magnitude and direction of associations may also differ by symptoms – as qualitative evidence points to the failure of some patients with chronic disorders which may lead them to see their doctor regularly, to recognise or articulate new symptoms. This points to health literacy as a driver in differences in stage at diagnosis.	Thank you for this comment. In the Discussion we have added text to our patient (behavioural) explanation of associations between pre-existing conditions and stage at diagnosis (4th paragraph): “Alternatively, the patient may not report potential cancer symptoms to their healthcare provider as they don’t see these symptoms as additional to their pre-existing condition, i.e. they do not recognise the seriousness of symptoms and / or are unaware of which symptoms relate to which disease.” We have also added further text to the Discussion (6th paragraph): “An inability to recognise and / or articulate new cancer-related symptoms to healthcare providers may be influential in associations between pre-existing mental

Comment received	How reflected and addressed
	health conditions and timeliness of cancer diagnosis. For example, a vignette study found that patients with Anxiety or Depression were more likely to attribute symptoms of colorectal cancer to mental health conditions rather than to cancer.”
[2] The rationale behind choice of lung and colon cancer is not quite clear – should be founded in scientifically sound reasons and not convenience. Please add this information to the background.	Thank you for this comment. We updated the Introduction section as follows (3rd paragraph): “In our study we chose to focus on lung and colon cancers. These are common cancers, frequently diagnosed at advanced stage and via emergency presentation, and having high mortality. ”
[3] The study period is only up to 2018 and thus not as up-to-date as would be optimal, especially with a growing focus on early diagnostics an updated dataset would have been preferred. In nature, registry-based data will not be completely up to date, but this study period ends 6 years ago.	Epidemiological studies by nature are always ‘out of date’. The key issue is however highlighted by the Reviewer’s comment is whether the underlying mechanisms and associations are likely to have been affected by the era. We posit that this is unlikely. The prevalence and management of both painful conditions and anxiety/depression is largely unchanged, as is the organisation of primary care services and community diagnostics such as chest x-ray. Furthermore, in view of the impact that the early years of the COVID-19 pandemic had on cancer diagnosis pathways in England (i.e. suspending cancer screening and delaying routine diagnostic investigations), we believe that including data from this period may bias our results, making interpretation of more recent eras more complex.
[4] For the reader, it may be helpful to add that the deprivation indicator is area-based	Thank you for this comment. The following text in bold has been added to the Methods section of the manuscript (‘Study Sample and Data Sources’ section, 2nd paragraph): “Registry data provided information on patient characteristics (age at diagnosis, sex and deprivation quintile of the Income domain of the Indices for Multiple Deprivation, an area-based measure of deprivation, based on the Lower layer Super Output Area (LSOA) – a geographic area of mean population 1,500 people - the patient resided in at the time of diagnosis) and diagnostic information (cancer site, date of diagnosis and stage of diagnosis).”
[5] The outcome is dichotomized but I would like a rationale for including stage III in the advanced stage group - could consider IIIb-IV for lung and IV for colon cancer? Does the chosen dichotomization align with the treatment protocols in UK during the study period?	Thank you for this comment. We acknowledge that stage IIIa colon and stage III lung cancer may be deemed non-advanced cancer from the point of view of management. In our analysis we defined early stage as stages I or II (including stage III in the advanced stage group) in line with the definition of early stage cancer in Government targets set out in England in the NHS Long Term Plan.³ We have, however, run a sensitivity analysis with advanced stage as stage IV only (versus stages I-III) to compare results (Supplementary Table 3) and found that the pattern in associations in these analyses were similar to those reported in our main analysis, with only some small variation in effect size.

Comment received	How reflected and addressed
	We have added the following text to the Methods (Statistical analysis section, 2nd paragraph): “We ran a separate sensitivity analysis, redefining early (stages I-III) and advanced (stage IV) stage, to compare with our main analysis results.” We have added the following text to the Results (‘Chronic Analgesic Medication and Anxiety / Depression as predictors of cancer stage at diagnosis’ section, 1st paragraph): “Similar associations were found in our sensitivity analysis, modelling stage as early (stages I-III) and advanced (stage IV) (Supplementary Table 4).”
[6] A large number of covariates were adjusted for in the multivariate regression models, and I assume these variables are all seen as confounders. I.e. a lot of the chronic disorders may include symptoms of pain - and thus the associations between painful conditions and number of morbidities or number of consultations may be complex - and some may act on mediators on same associations? Please clarify by presenting a directed acyclic graph to justify selection of confounders - and refrain from adjusting for any mediators.	Thank you for this comment. We now include a Directed Acyclic Graph (DAG) (Supplementary Figure 2), showing the associations between the covariates we included in our model, the pre-existing conditions (Chronic Analgesic Medication and Anxiety / Depression) and stage at cancer diagnosis. As reflected in the DAG, we do not consider any of the covariates to be on the causal pathway or mediators in our analysis. We have added the following sentence to the Methods section of the manuscript (‘Study variables’ section, 5th paragraph): “We illustrated our hypothesised relationships between Chronic Analgesic Medication or Anxiety / Depression, covariates and stage at cancer diagnosis in a Directed Acyclic Graph (Supplementary Figure 2), using DAGitty software”
[7] Results: When reading the detailed and informative results, one thing may have been elaborated on - do we know if there was an association between painful conditions and acute presentation of colon cancer - which may often have a high stage (not always) but have a very high mortality. Would it be possible to include information on this in the dataset, to explore this further? Else, please include as a limitation as they usually amount to some 10% of the bowel cancer group.	Thank you for this comment. We did not adjust for emergency presentation in our analysis as emergency presentation is highly correlated with cancer stage, and we considered this to be a mediator in the association between pre-existing comorbidity and stage at cancer diagnosis. We are currently exploring associations between Chronic Analgesic Medication and Anxiety / Depression and emergency presentation as a separate piece of work.
[8] Discussion: The suggestion of 3 tentative explanations is bold and well argued - I suggest authors consider to include that the patients may not see symptoms as additional to their condition - as they may not know what is a symptom of what disease - this is relevant addition to the second (patient factors association).	Thank you for this suggestion. We have now incorporated this into the second explanation in the Discussion (added text is in bold) (4th paragraph): “ii) Patients are distracted from reporting additional symptoms to their healthcare provider, either because they are primarily focused on dealing with their recent-onset condition (competing demands, influencing help-seeking for cancer symptoms) or because they fear

Comment received	How reflected and addressed
	that they may be perceived as hypochondriac. Alternatively, the patient may not report potential cancer symptoms to their healthcare provider as they don't see these symptoms as additional to their pre-existing condition, i.e. they do not recognise the seriousness of symptoms and / or are unaware of which symptoms relate to which disease."
[9] Separating depression and anxiety based on prescription medication is not clear-cut but the discussion of results of the subanalyses is cautious. May results on anxiety indicate that symptoms of advanced lung or colon cancer actually may in themselves cause anxiety (dyspnea, stomach pain).	Thank you for this comment. We agree that there is a possibility of reverse causality, and that anxiety may result from symptoms of cancer. We have therefore reworded the text in the Discussion to clarify this (6th paragraph): "The underlying reason behind the association between recent-onset Anxiety and advanced stage cancer diagnosis is less apparent, but may be influenced by reverse causality: symptoms of advanced lung or colon cancer such as dyspnoea or abdominal pain, respectively, may cause anxiety."

Reviewer 4

Comment received	How reflected and addressed
This observational study examines the association between stage at diagnosis in colon and lung cancer patients and the type and duration of physical and mental health morbidities. These comorbid conditions were categorized as "recent onset" (< 12 months pre-cancer diagnosis), "historic" (12-72 months pre-cancer diagnosis), and "persistent" (< 12 and 12-72 months pre-diagnosis). The analysis identified the most prevalent conditions to be anxiety/depression and painful conditions; and findings from the multivariable regression analysis showed that for each of these conditions, recent onset was associated with higher odds of being diagnosed with late-stage colon or lung cancer. Conversely, the presence of these conditions were associated with lower odds of being diagnosed with late-stage disease when they were persistent because patients with persistent conditions may have greater healthcare utilization, and therefore greater opportunities to be diagnosed with earlier-stage disease. Prior studies have shown mixed results between the presence of comorbid conditions and stage at diagnosis; but the innovative feature of this study is in distinguishing between longstanding and recent comorbidities. The analysis is straight-forward, and the manuscript is well written. However, there are some major and other weaknesses:	Thank you for the apt summary of the key features and findings of our study, and its assessment of its position in the literature, and your very constructive comments, which we have engaged with as following:
Major weakness: As in most studies, comorbid conditions are dealt with one at a time; however, comorbidities often present in combination, rather than isolation--or as part of	Thank you for this comment. We agree that comorbidities are commonly present in combination. In the first part of our analysis, we used primary care data to derive the prevalence of 35 pre-existing physical and mental health conditions among the lung and

Comment received	How reflected and addressed
multimorbidity. Such combinations are not taken into account in the analysis.	colon cancer patients (Supplementary Table 2). Following this, in our multivariable analysis, we account for multimorbidity among the patients by including a variable denoting the number of pre-existing physical conditions patients have. We also provide information on distribution of variables describing number of physical conditions and number of mental health conditions by stage at diagnosis in the Characteristics tables (Table 1) .
1- The study uses data from the England National Cancer Registry linked with the Clinical Practice Research Datalink, CPRD data. The CPRD data set provides data for a sample of 6.9% of the UK population. This is a fairly small sample of the population. How confident are the authors that they can generalize their findings to the population overall?	Thank you for this comment. While 6.9% of the UK population may be considered a small sample, CPRD is broadly representative of the UK population in terms of age, sex and ethnicity.⁴ This supports the generalisability of our findings to the overall population. Additionally, the use of CPRD data is wide-reaching. Over 3,500 published peer-reviewed CPRD studies are part of an evidence base used to inform UK clinical guidance and best practice.⁵
2- What do 'readcode' and 'prodcode' stand for? I am not familiar with these terms.	Thank you for this comment. As per our response to Reviewer 1, we have included the following text in the Methods section of the manuscript ('Study variables' section, 2nd paragraph): "We derived information on 35 pre-existing physical and mental health comorbidities from GP consultation records and prescription data using 'readcode' and 'prodcode' definitions from the Cambridge Multimorbidity Score (CMS) (Supplementary Table 1). 'Readcode' is a data field in CPRD to record patient findings and procedures, based on codes used within the National Health Service, while 'prodcode' is a CPRD data field representing product codes describing drug and appliance prescriptions."
3- What is the rationale for selecting colon and lung cancer for the analysis? Why were other cancers left out?	As per our response to Reviewer 3, we focused on colon and lung cancer as they are among the most common cancers, are frequently diagnosed at advanced stage and via emergency presentation and have high mortality. We think it would be interesting to conduct further research to explore how timing and duration of these physical and mental health morbidities influence stage at diagnosis of other cancers, and have added a sentence to this effect in the Discussion (last paragraph): "Furthermore, it would be informative to conduct further research to understand how timing and duration of these physical and mental health morbidities may influence stage at diagnosis of other cancers."
4- The authors state that they decided to focus on painful conditions and depression/anxiety because they were the	Thank you for this comment. We agree that there are other prevalent pre-existing conditions, some mostly historic conditions and not recorded < 12 months before cancer

Comment received	How reflected and addressed
most prevalent conditions; however, based on the data presented in Supplemental Table 2, there are several other conditions that are prevalent, especially in the "historic" and "persistent" categories. The authors should therefore find a better justification for focusing on painful conditions and depression/anxiety.	diagnosis. We chose to focus on conditions that were prevalent in the 12 months prior to cancer diagnosis on the basis that conditions present in this period may be most likely to influence pre-cancer patterns in healthcare use, diagnostic investigations and timely diagnosis of cancer.^{1, 6} In our analysis, we account for the presence of multimorbidity by including a variable denoting number of other pre-existing conditions. Further research could explore the role of other specific conditions on stage at cancer diagnosis.
5- In the multivariable analysis, could there be an issue of collinearity between the number of visits and the categories of "recent", "historic", and "persistent"?	Thank you for this comment. We checked for collinearity between the categorical variables (i.e. Number of GP visits against Chronic Analgesic Medication [Painful Conditions] and against Anxiety / Depression) using Cramer's V test. The test result was <0.2 for both Chronic Analgesic Medication and Anxiety / Depression, indicating only a small association between variables.⁷

REFERENCES

1. Renzi C, Kaushal A, Emery J, Hamilton W, Neal RD, Rachet B, et al. Comorbid chronic diseases and cancer diagnosis: disease-specific effects and underlying mechanisms. *Nat Rev Clin Oncol*. 2019;16(12):746-61.
2. Gurney J, Sarfati D, Stanley J. The impact of patient comorbidity on cancer stage at diagnosis. *Br J Cancer*. 2015;113(9):1375-80.
3. NHS England. NHS Long Term Plan 2019 [Available from: <https://www.longtermplan.nhs.uk>].
4. Herrett E, Gallagher AM, Bhaskaran K, Forbes H, Mathur R, van Staa T, et al. Data Resource Profile: Clinical Practice Research Datalink (CPRD). *International Journal of Epidemiology*. 2015;44(3):827-36.
5. Medicines & Healthcare products Regulatory Agency. Clinical Practice Research Datalink 2025 [Available from: <https://www.cprd.com/#:~:text=For%20more%20than%2035%20years,delivery%20and%20disease%20risk%20factors>].
6. White B, Renzi C, Rafiq M, Abel GA, Jensen H, Lyratzopoulos G. Does changing healthcare use signal opportunities for earlier detection of cancer? A review of studies using information from electronic patient records. *Cancer Epidemiology*. 2022;76:102072.
7. Cohen J. *Statistical Power Analysis for the Behavioral Sciences*. 2nd ed. New York: Routledge; 1988.

Response to Reviewer 4

Comment	Relevant section	Response
The authors have satisfactorily addressed the most of the comments by reviewers. However, to address the comment on multimorbidity by Reviewer 4, the authors have simply included the number of physical and mental comorbidities in the multivariable models. While this is helpful, the insight gained by looking at combinations of comorbid conditions would be far richer and more informative than accounting for the number of chronic conditions. This may be outside the scope of this study. At a minimum, however, this should be noted in the limitations, and suggested as a future study.	Discussion – ‘limitations’ paragraph.	Thank you for prompting us to provide further reasoning on this issue. We agree that count measures of morbidities, as used in our study, do not account for the fact that different combinations of morbidities will have different associations with the outcome of interest. However, the total number of combinations of morbidities is very large. For example, for the 19 conditions conventionally included in the Charlson morbidity score, and considering pairwise combinations only, there are 171 such combinations, while the total number of all possible combinations exceeds half a million. To simplify matters, and concordant it most of literature in the field, our Group tends to use count measures of morbidity, though occasionally we have also adjusted for single specific major morbidities. Given that the principal research question of our study does not directly relate to potential confounding by combinations of other morbidities, as suggested by the Reviewer, we have added a substantial new paragraph in Discussion, Limitations to discuss the issue within the context of our study and suggesting further work. We include the relevant section below, with the new text denoted in bold fonts below: We acknowledge some limitations with our study. First, approximately 10% of lung cancer patients and 15% of colon cancer patients had missing information on stage at diagnosis, our outcome of interest. We excluded these patients from our multivariable analyses given the small percentage of missingness, and because other methods of handling missing data, such as multiple imputation, have been shown to be less efficient when, as in our analysis, missing data relate to the outcome variable³⁵. Second, the presence of mental health conditions may be underestimated in administrative health data. To mitigate this, we captured relevant information on mental health conditions from both primary care consultation and prescription data. Third, we have adjusted for multimorbidity by using a simple count of conditions. In theory it may be more attractive to consider adjusting for specific combinations of conditions, but in reality this is complex. For example, for 19 specific conditions conventionally included in the Charlson Comorbidity Index³⁶, there are 171 pairwise combinations, and over half a million possible combinations overall. Adjusting for a small number of specific conditions, selected on the basis they may influence the diagnostic process of cancer, may be an alternative solution when adjusting for multimorbidity in future work. Using colon cancer diagnosis as an example,

Comment	Relevant section	Response
		this would include conditions such as history of Cardiovascular Disease, previously shown to reduce the likelihood of receiving colonoscopy for investigating cancer symptoms, ³⁷ or Inflammatory Bowel Disease, a condition providing an 'alternative explanation' for colon cancer symptoms, which has been shown to delay cancer diagnosis. ³⁸ "